# Life and death in the Chicxulub impact crater: A record of the Paleocene-Eocene Thermal Maximum

Vann Smith[1,2], Sophie Warny[1,2], Kliti Grice[3], Bettina Schaefer[3], Michael T. Whalen[4], Johan Vellekoop[5,6], Elise Chenot[7], Sean P.S. Gulick[8,9,10], Ignacio Arenillas[11], Jose A. Arz[11], Thorsten Bauersachs[12], Timothy Bralower[13], François Demory[14], Jérôme Gattacceca[14], Heather Jones[13], Johanna Lofi[15], Christopher M. Lowery[9], Joanna Morgan[16], Noelia B. Nuñez Otaño[17], Jennifer M.K. O'Keefe[18], Katherine O'Malley[4], Francisco J. Rodríguez-Tovar[19], Lorenz Schwark[3,12], and the Expedition 364 Scientists

[1]Department of Geology and Geophysics, Louisiana State University, Baton Rouge, LA 70803, USA
[2]Museum of Natural Science, Louisiana State University, Baton Rouge, LA 70803, USA
[3]Western Australian Organic and Isotope Geochemistry Centre, The Institute for Geoscience Research, School of Earth and Planetary Science, Curtin University, Perth, WA 6102, Australia
[4]Department of Geosciences, University of Alaska Fairbanks, Fairbanks, AK 99775, USA
[5]Department of Earth and Environmental Sciences, Division of Geology, KU Leuven, 3001 Heverlee, Belgium
[6]Analytical, Environmental and Geo-Chemistry (AMGC), Vrije Universiteit Brussel, 1050 Brussels, Belgium
[7]GeoRessources, Université de Lorraine, CNRS, 54 500 Vandœuvre-lès-Nancy, France
[8]Department of Geological Sciences, Jackson School of Geosciences, University of Texas at Austin, TX 78712, USA
[9]Institute for Geophysics, Jackson School of Geosciences, University of Texas at Austin, TX 78712, USA
[10]Center for Planetary Systems Habitability, University of Texas at Austin, TX 78712, USA
[11]Departamento de Ciencias de la Tierra e Instituto Universitario de Investigación de Ciencias Ambientales de Aragón, Universidad de Zaragoza, Pedro Cerbuna 12, E-50009 Zaragoza, Spain
[12]Department of Organic Geochemistry, Institute of Geosciences, Christian-Albrechts-University, Kiel, 24118, Germany
[13]Department of Geosciences, Pennsylvania State University, University Park, PA 16801, USA
[14]CNRS, Aix-Marseille Univ, IRD, Coll France, INRAE, CEREGE, Aix-en-Provence, France
[15]Géosciences Montpellier, 'Université Montpellier, CNRS, Montpellier, France
[16]Department of Earth Science and Engineering, Imperial College London, SW7 2AZ, UK
[17]Facultad de Ciencia y Tecnología (FCyT), Universidad Autónoma de Entre Ríos, CONICET, Laboratorio de Geología del Neógeno-Cuaternario, Diamante, Entre Ríos, Argentina
[18]Department of Physics, Earth Science, and Space Systems Engineering, Morehead State University, Morehead, KY, USA
[19]Departamento de Estratigrafía y Paleontología, Facultad de Ciencias, Universidad de Granada, 18002 Granada, Spain

*Correspondence to*: Vann Smith (vannpaleo@gmail.com)

**Abstract.** Thermal stress on the biosphere during the extreme warmth of the Paleocene-Eocene Thermal Maximum (PETM) was most severe at low latitudes, with sea surface temperatures at some localities exceeding the 35° C at which marine organisms experience heat stress. Relatively few equivalent terrestrial sections have been identified, and the response of land plants to this extreme heat is still poorly understood. Here, we present a new record of the PETM from the peak ring of the Chicxulub impact crater that has been identified based on nannofossil biostratigraphy, an acme of the dinoflagellate genus *Apectodinium*, and a negative carbon isotope excursion. Geochemical and microfossil proxies show that the PETM is marked by elevated $TEX_{86}^H$-based sea surface temperatures (SSTs) averaging ~37.8 °C, an increase in terrestrial input and surface productivity, salinity stratification, and bottom water anoxia, with biomarkers for green and purple sulfur bacteria indicative

of photic zone euxinia in the early part of the event. Pollen and plants spores in this core provide the first PETM floral assemblage described from México, Central America, and the northern Caribbean. The source area was a diverse coastal shrubby tropical forest, with a remarkably high abundance of fungal spores indicating humid conditions. Thus, while seafloor anoxia devastated the benthic marine biota, and dinoflagellate assemblages were heat-stressed, the terrestrial plant ecosystem thrived.

## 1 Introduction and geologic setting

The Paleocene-Eocene Thermal Maximum (PETM) was a period of global warming associated with ocean acidification, an intensified hydrological cycle, reduction in marine dissolved oxygen concentrations, eustatic sea level rise, and major ecological shifts (e.g., Zachos et al., 2003; Gingerich 2006; Dickson et al., 2014; Sluijs et al., 2008; Carmichael et al., 2017). Recent age estimates place the PETM at approximately 55.93-55.71 Ma (Westerhold et al. 2017; Hollis et al. 2019). The onset of the PETM is marked by a global negative carbon isotope excursion (CIE) (Dickens et al. 1997; Gradstein et al. 2012). Possible sources of this isotopically light carbon include methane clathrates, combustion of organic matter, thermogenic methane, and organic matter released from permafrost (e.g., McInerney and Wing, 2011). Sea surface temperature (SST) during the PETM in some low-latitude regions exceeded 35 °C, resulting in heat stress for eukaryotic plankton (e.g., Frieling et al., 2018). In contrast, the few existing PETM records of low-latitude terrestrial plant assemblages indicate an increase in diversity (e.g., Jaramillo et al., 2010; Srivastava and Prasad, 2015; Prasad et al., 2018). Here, we established a new multiproxy record of the response of marine and terrestrial biota to the PETM in the western Caribbean/Gulf of Mexico at International Ocean Discovery Program (IODP) Expedition 364 Site M0077. This record includes the first published pollen and spore PETM assemblage from tropical North America (Smith et al., 2020a, 2020b). These data allow us to determine the extent of marine and terrestrial heat stress from the understudied region and determine how they compare with other PETM sections.

International Ocean Discovery Program (IODP)-International Continental Scientific Drilling Program (ICDP) Site M0077 is located on the peak ring of the Chicxulub impact crater in the Yucatán Peninsula, México (Fig. 1) (Morgan et al., 2017). The crater was a marine basin in the Paleogene, with mainly pelagic and outer-platform sediment deposition (Lefticariu et al., 2006). Immediately after impact, some of the rim may have been subaerially emergent (Morgan et al., 1997), but, if so, would have been quickly eroded. During the PETM, only isolated areas of the crater rim may have been emergent, given the existence of an embayment into the crater to the north and northeast (Gulick et al., 2008). Although PETM records from the Gulf of Mexico are scarce, another site in the Chicxulub crater, the Yaxcopoil-1 (Yax-1) core, contains a PETM section identified by a negative carbon isotope excursion, deposited during a period of maximum flooding (Whalen et al., 2013) (Fig. 1). The PETM has also been identified on the Mississippi paleoshelf (Fig. 1), where evidence indicates increased $TEX_{86}^{H}$-based SSTs, photic zone euxinia, and sea level rise (Sluijs et al., 2014).

## 2 Methods

Quantitative palynological abundances are expressed in terms of specimens per gram, using a *Lycopodium* spike. Species counts, descriptions, and paleoecological interpretations can be found in Smith et al. (2020a, 2020b). The D/S ratio between dinoflagellate cysts and pollen/plant spores is described in Warny et al. (2003). The degree of bioturbation has been quantified using the Bioturbation Index (BI) (Taylor and Goldring, 1993), a descriptive classification ranging from 0 (no bioturbation) to 6 (completely bioturbated). Samples for $\delta^{15}N$ and $\delta^{13}C_{TOC}$ (n = 65) analyses were prepared by acidifying approximately 0.5 g powdered material with an excess of 1 M HCl. The acid-insoluble residues were neutralized, freeze-dried and analyzed for their carbon and nitrogen contents as well as stable isotope compositions using a Costech Elemental Analyzer (ECS 4010) and a Delta+XP mass spectrometer. Typical instrumental precision of the isotope measurements is <0.2‰. $\delta^{15}N$ is reported relative to atmospheric $N_2$ and $\delta^{13}C_{TOC}$ is reported relative to VPDB. Clay mineral assemblages were identified by X-ray diffraction on oriented mounts of non-calcareous clay-sized particles (<2 μm). SSTs based on isoprenoidal glyceroldialkylglyceroltetraethers (isoGDGTs) (Schouten et al. 2002) were reconstructed using the $TEX_{86}^{H}$ calibration of Kim et al. (2010). Palynological sampling resolution is approximately 5 cm, and $\delta^{15}N$ and $\delta^{13}C_{TOC}$ sampling resolution approximately 0.5 cm, in the body of the PETM section. Biomarker analysis in the Late Paleocene section was hampered by low TOC, with only one sample suitable for $TEX_{86}$ measurement. Generally, the sampling strategy was designed for high resolution analysis of the body of the PETM section, which appears to be bounded at the top and bottom by unconformities. Additional methods are provided as supplementary materials along with all data.

## 3 Results

IODP-ICDP drilling at Site M0077 recovered Paleocene to early Eocene post-impact sedimentary rocks between 617.33-505.70 meters below seafloor (mbsf). The PETM section (607.27-607.06 mbsf, Core 37R-1) is a laminated black to dark gray shale, separated from an upper Paleocene carbonate hardground by an unconformity, and unconformably underlying a burrowed lower Eocene packstone at the top of Core 37R-1 and through Core 36R-4 (Fig. 2). The uppermost Paleocene, underlying the PETM interval, is characterized by two significant disconformities. The lower disconformity is atop a 6-8 cm thick gray claystone (607.68 mbsf), interpreted to be a bentonite, with an erosionally scoured upper surface. It is overlain by a 7 cm thick carbonate rudstone (607.68-607.61 mbsf) that grades upward into a 22 cm thick packstone (607.61-607.39 mbsf). The rudstone contains claystone and carbonate lithoclasts up two 2 cm in diameter, foraminifera, and lime mud, and grades into a light gray foraminiferal packstone with wispy stylolitic laminae. The packstone is overlain by a ~4 cm thick gray claystone (607.39-607.35 mbsf). Both the contact between the packstone and claystone and the claystone itself are burrowed and one burrow is infilled by material from the overlying facies. The claystone is abruptly overlain by a carbonate grainstone (607.32-607.27 mbsf) with planktic and large benthic foraminifera, red algae, ostracods, calcispheres, and black and gray carbonate lithoclasts. The top of this unit (607.27 mbsf) is a hardground and disconformity with about 1 cm of relief which separates the Paleocene and PETM sections. The lower contact of the grainstone with the underlying

claystone (607.32 mbsf) also appears to be unconformable, but no biozones are missing, so it may represent a diastem rather than a significant hiatus.

The PETM interval (607.27-607.06 mbsf) is about 21 cm thick. It has a sharp basal contact that drapes over the relief atop the underlying hardground. The PETM interval consists of dark gray to black shale that is laminated at the mm scale. The base of the interval is slightly lighter colored gray shale and contains clay, organic matter, sand-sized carbonate lithoclasts and foraminifera eroded from the underlying unit, as well as rare green grains and spherules that appear to be altered impact glass. The remainder of the PETM interval consists of mm-scale laminae that are usually dark gray at their base and black at the top and contain quartz, muscovite, rare plagioclase silt grains, and rare calcispheres. Laminae are commonly defined at their base by quartz and muscovite silt and grade upward into clay and organic-rich shale. The uppermost PETM shale is bioturbated, with burrows infilled with material from the overlying carbonate packstone. The interval directly overlying the PETM (607.06-606.85 mbsf) also contains abundant reworked material, including several pebble-sized clasts of limestone which appear to contain Cretaceous foraminifera. Above the core gap, Cores 36R-3 and 36R-2 are composed of a pale massive packstone with two black chert layers, at 606.62-606.56 and 606.16-606.11 mbsf (Figure 2).

Bioturbation is absent to minimal in the PETM, with rare *Chondrites* ichnofossils, except at the top of the interval (607.11-607.06 mbsf) where *Planolites* burrows are observed, infilled with sediment from the overlying packstone. The clay mineral assemblages are dominated by R0 random illite/smectite mixed layers (up to 90%), and also contains traces of chlorite, illite, and palygorskite. The latter is rare in the upper Paleocene, and increases in abundance through the PETM, reaching a peak of 5% relative abundance at 607.08 mbsf. The PETM interval is characterized by a marked increase in magnetic susceptibility ($\chi$), anhysteretic remanent magnetization (ARM), and isothermal remanent magnetization (IRM). The average values increase by a factor of 15.7, 5.8, and 12.4 for $\chi$, ARM, and IRM, respectively, compared to the average values over the analyzed pre-PETM interval (607.67-607.27 mbsf) (see supplementary materials).

Total organic carbon (TOC) is low above and below the PETM (Fig. 2), with high concentrations (>6% rock weight) in the upper PETM section. Total organic carbon/total nitrogen (TOC/TN) ratios (e.g., Meyers and Shaw, 1996) range from 0.6 to 6.8 in the upper Paleocene, with higher values averaging 10.7 in the PETM section. TOC/TN values in the post-PETM section range from 1.4 to 4.7. $\delta^{13}C_{TOC}$ (total organic carbon $\delta^{13}C$) ranges from -27.5‰ to -25.8‰ in the upper Paleocene and is -28.4‰ at the base of the PETM section, generally becoming more negative upsection through the PETM, with the most depleted value of -30.1‰ in the upper PETM (607.12 mbsf). Above 607.07 mbsf, $\delta^{13}C_{TOC}$ values become more positive, then stabilize at -27.5‰ at 607.03 mbsf. $\delta^{15}N$ ranges from 1.0‰ to 3.7‰ in the upper Paleocene and is 5.3‰ at the base of the PETM section, with more depleted values through the PETM, reaching a minimum of -2.0‰ at 607.21 mbsf. The PETM $\delta^{15}N$ record is marked by two negative excursions with values below 0‰, separated by a brief interval of positive $\delta^{15}N$ values between 607.17-607.13 mbsf. Above 607.10 mbsf, $\delta^{15}N$ values become more positive, with a value of 0.9‰ at 607.02 mbsf (Fig. 2).

TEX$_{86}^H$-based SSTs and other biomarkers were difficult to retrieve in the Late Paleocene due to low organic matter content (TOC values often <0.1%), but a single sample at 607.33 mbsf yielded a TEX$_{86}^H$-based SST of 34.0 °C. In the PETM interval, TEX$_{86}^H$-based SSTs ranged from 37.4-38.0 °C, averaging 37.8 °C. Just above the PETM section, at 607.05 mbsf, the TEX$_{86}^H$-based SST was 37.9 °C, followed by a decrease in SSTs to 37.1 °C and 37.3 °C at 606.87 and 606.72 mbsf, respectively (Fig. 2). To verify the applicability of the TEX$_{86}$ proxy a series of complementary molecular indicators, the BIT (Branched and Isoprenoid Tetraether) index (Hopmans et al., 2004), MI (Methane Index), and f$_{cren}$ (relative abundance of the crenarchaeol regio-isomer) were calculated, all of which passed the exclusion criteria as summarized in O'Brien et al. (2017). Green and purple sulfur bacteria biomarkers (chlorobactane, isorenieratane and okenane) reach their highest concentrations near the bottom of the PETM section, with low concentrations through the rest of the event (Fig. 2).

Nannofossil abundances decrease through the PETM section and become rare in the post-PETM section. Foraminifera at Site M0077 are abundant in the upper Paleocene section but are absent to very rare in the PETM section, with evidence of reworking. Dinosteranes, biomarkers associated with dinoflagellates (e.g., Summons et al., 1987) have relatively high concentrations in the upper Paleocene and lower PETM section, with decreased abundance in the PETM and post-PETM sections. Organic-walled microfossils are absent to rare in the Paleocene. Dinoflagellate cyst concentrations peak at 607.26 mbsf, with a decreasing trend through the rest of the PETM (Fig. 2). Relative abundances of *Apectodinium* are highest at 607.26 mbsf, and decrease through the PETM, while the highest relative abundances of Goniodomidae are found just above the event. Fungal spore concentrations peak in the middle of the PETM section (Fig. 2), reaching concentrations much higher (>400 specimens/gram) than any other samples, including samples with higher overall palynomorph concentration and excellent preservation in the later Ypresian section near the top of the core (520.79-505.88 mbsf), indicating the fungal spike is not a taphonomic artifact. The PETM fungal assemblage is dominated by *Nigrospora*-types, which are common leaf endophytes on a variety of substrates, including soil, and are commonly airborne (Wang et al. 2017). The PETM pollen and plant spore assemblage is dominated by *Malvacipollis* (Euphorbiaceae), *Ulmipollenites* (Ulmaceae), *Bohlensipollis* (Eleagnaceae), and angiosperm pollen of unknown lower botanical affinity, with rare gymnosperm pollen and lower plant spores.

## 4 Discussion

### 4.1 Stratigraphy and depositional environment

As described earlier, the PETM section in the Site M0077 core is bracketed by unconformities and incomplete, with the onset and recovery missing, and only part of the PETM section is preserved. The fine-grained nature and lack of sedimentary structures indicating current deposition indicate that the PETM interval recovered was deposited in relatively deep, quiet water with sediments largely settling from suspension. The laminated black shale points toward low oxygen conditions. However, the trace fossil assemblage implies that anoxia and/or euxinia were likely intermittent. Water depths for Site M0077 during most of the Paleocene were on the order of 600-700 m (Lowery et al., 2018) but the facies

immediately underlying and overlying the PETM interval contain numerous grains from shallow water environments, like larger benthic foraminifera and red algae which indicate either relatively shallow water in the crater or extensive reworking from the crater margin. Assigning a water depth for the PETM interval is complicated by the complete lack of obviously in situ depth-sensitive benthic foraminifera that could provide such insight. However, the presence of deeper dwelling planktic foraminifera such as *Subbotina* spp. and *Globanomalina pseudomenardii*, which occupied a thermocline habitat (e.g., Aze et al., 2011), indicate that the water was at least deep enough for the establishment of stratification. The PETM is globally characterized by an eustatic sea level rise (Sluijs et al., 2008) so water depths were likely somewhat deeper during the PETM than during the times when the units above and below were deposited. The reworking observed in PETM age sediments in the Yacopoil-1 core (Whalen et al., 2013) suggests that reworking in the Chicxulub crater was common during the PETM and the shallow water foraminifera observed in the PETM section at Site M0077A were likely reworked from the crater rim.

The PETM age of the shale interval at Site M0077 (607.27-607.06 mbsf) has been confirmed by a negative carbon isotope excursion (CIE) and biostratigraphy. The earliest nannofossil PETM sample, at 607.25 mbsf, contains *Discoaster salisburgensis* var. *anartios*, a characteristic PETM excursion taxon (e.g., Bralower and Self-Trail, 2016). The global negative CIE is also observed at Site M0077 (Fig. 2). In complete records of the PETM, the peak of the negative CIE and highest temperatures are observed within the first ~20 ky of the event, followed by a gradual recovery to more positive $\delta^{13}C_{TOC}$ values and lower SSTs (e.g., Hollis et al., 2019). However, at Site M0077, the most depleted $\delta^{13}C_{TOC}$ values are found in the upper part of the interval. The onset and peak of the PETM CIE are thus missing due to erosion or non-deposition. The abrupt shift to more positive $\delta^{13}C_{TOC}$ values at 607.06 mbsf suggests that the later PETM section and immediate recovery is also missing, with another unconformity at the top of the PETM section. The trend towards more negative $\delta^{13}C_{TOC}$ values in the PETM can be explained as the result of an increasing contribution of terrestrial organic matter. This explanation is consistent with the palynological D/S ratio, which shows the highest relative abundance of terrestrial versus marine palynomorphs at approximately the same depth as the most negative $\delta^{13}C_{TOC}$ values (Fig. 2). Increasing TOC/TN ratios are also consistent with a higher input of terrestrial organic matter through the PETM (e.g., Burdige, 2006). Lithologically, the PETM section is clearly distinguished from the Paleocene section by an abrupt switch from carbonate to siliciclastic clay deposition, and an abrupt increase in detrital input, as indicated by increased magnetic parameters.

### 4.2 PETM environmental change

SSTs were estimated using the relative abundance of thaumarchaeotal isoGDGTs. We here used the $TEX_{86}^H$ SST-calibration of Kim et al. (2010) developed for the determination of SSTs in (sub)tropical oceans and low latitude settings. The uncertainties associated with $TEX_{86}$ estimates of SSTs exceeding the present-day SST maximum of 27-29 °C have been addressed for Cretaceous (O´Brien et al., 2017) and Paleogene (Frieling et al., 2017) strata. These authors conclude that during hyperthermals $TEX_{86}^H$ delivers reliable SST reconstructions, with an upper calibration limit occurring at 38.6 °C (O`Brien et al., 2017). The $TEX_{86}$ ratios in the PETM section approach unity (0.96-0.98), nearing the theoretical upper limit for temperature reconstruction using this proxy. BAYSPAR (Tierney and Tingley, 2014) and linear (Schouten et al., 2007)

TEX$_{86}$ calibrations yield extremely high PETM SSTs in excess of 44 °C, above the heat tolerance for most dinoflagellates, foraminifera, and other eukaryotic plankton. GDGT abundance data are provided in the supplementary material so that alternative and possible future TEX$_{86}$ calibrations can be applied to the dataset. In previous studies potential impacts on the TEX$_{86}$ proxy have been identified and a series of validation criteria developed as summarized in O´Brien et al. (2017). Application of these validation proxies identified all samples to fulfil the exclusion criteria for the use of the TEX$_{86}^{H}$ paleothermometer. The thermal maturity as determined by the side chain isomerization of the C$_{29}\alpha\alpha\alpha$ steranes [20S/(20S+20R)] and C$_{31}\alpha\beta$ hopanes [20S/(20S+20R)] is 0.13, and 0.34, respectively (see supplementary materials), which is indicative of a low maturity, equivalent to a vitrinite reflectivity of 0.30-0.35%. This is supported by Rock Eval T$_{max}$ values averaging 428°C. A maturity impact on the GDGT data is thus considered to be minimal and affecting all samples to an equal extent. Preservation of immature biomarkers is further supported by presence of thermally labile aromatic carotenoids.

TEX$_{86}^{H}$-based SSTs increased by ~4 °C between the Late Paleocene and PETM (Fig. 2), with average PETM SSTs determined here of 37.8 °C, similar to values observed in the eastern equatorial Atlantic (Frieling et al., 2018) and the Dahomey Basin, western Africa (Frieling et al., 2017), and ~3 °C higher than those observed in the Harrell core (Sluijs et al., 2014) on the northern Gulf of Mexico Margin and Wilson Lake core (Zachos et al., 2006) on the mid-Atlantic North American margin. Jaramillo et al. (2010) estimated Late Paleocene SSTs of 28-31 °C and Early Eocene SSTs of 31-34°C from Colombia using TEX$_{86}$ measurements, although no PETM age TEX$_{86}$ measurements were available. Frieling et al. (2017), investigating a tropical marine PETM record from Nigeria, estimated latest Paleocene SSTs of 32-34 °C, with average PETM SSTs of ~36 °C. The temperature increase from the Late Paleocene to PETM section at Site M0077A is consistent with estimates of a 4-5 °C global mean surface temperature anomaly for the PETM (Dunkley Jones et al., 2013). SSTs decrease to 37.1 and 37.3 °C in the post-PETM section at 606.87 and 606.72 mbsf, respectively.

Several lines of evidence indicate increased terrestrial input during the PETM, including increased concentrations of terrestrial palynomorphs, increased D/S and TOC/TN ratios, and an increase in detrital ferromagnetic minerals. Theoretically, this increase in terrestrial input could be the result of a relative sea level fall, but this would not be consistent with an interpreted PETM sea level rise in the Gulf of Mexico and globally (Sluijs et al., 2014). Instead, the increase in terrestrial input is interpreted to result from an intensified hydrological cycle during the PETM, as noted in other studies (e.g., Crouch et al., 2003; Bowen et al., 2004; Schmitz and Pujalte, 2007; Handley et al., 2012). The exceptionally high abundance of fungal spores in the PETM section suggests that increased humidity and terrestrial weathering resulted in greater detrital and nutrient input to Site M0077. BIT index values, which have been used as a proxy for terrestrial organic matter in sediments (Hopmans et al., 2004; Weijers et al., 2006), are higher in the Late Paleocene than in the PETM section (Fig. 2). Low BIT values in samples from the PETM section may indicate a source of terrestrial organic matter lean in soil microbial matter (Huguet et al., 2007; Schouten et al., 2013), possibly from low-lying carbonate terrain to the south (Fig. 1) and/or an increased productivity of Thaumarchaeota.

The relative abundance of the clay mineral palygorskite increases through the PETM section. Higher abundances of palygorskite in other PETM sediments have been interpreted as evidence for increased aridity (Carmichael et al., 2017), as

palygorskite commonly forms in coastal marine environments where continental alkaline waters are concentrated by evaporation (Bolle and Adatte, 2001). At Site M0077, the palygorskite may have originally formed in hypersaline lagoonal environments similar to other Eocene-Oligocene palygorskite deposits in the Yucatán Peninsula (de Pablo Galán, 1996). The increase in relative abundance of palygorskite through the PETM section may therefore be the result of increased fluvial transport of sediments to Site M0077 from lagoonal environments to the south, rather than the result of increased aridity.

The near absence of bioturbation in the PETM section, with preserved sedimentary lamination and high TOC, is consistent with bottom water anoxia through much of the PETM, and sulfur bacteria biomarkers are indicative of photic zone euxinia (e.g., Summons and Powell, 1987; Grice et al., 2005; Sluijs et al., 2014) in the earlier PETM record. Depleted $\delta^{15}N$ values similar to those observed during ocean anoxic events (e.g., Jenkyns, 2010) can be explained by upwelling of ammonium from anoxic deep waters during periods of high nutrient availability (e.g., Higgins et al., 2012), or increased

cyanobacterial $N_2$ fixation (e.g., Bauersachs et al., 2009). The transient positive $\delta^{15}N$ excursion in the middle of the PETM section at Site M0077 (Fig. 2) is similar to the $\delta^{15}N$ PETM record of Junium et al. (2018) from the northern Peri-Tethys seaway, with depleted $\delta^{15}N$ in the top and bottom of the PETM section, separated by an interval of more enriched $\delta^{15}N$, which they interpreted to result from a more oxic, less stratified water column, possibly due to reduced freshwater influx.

**4.3 Implications for life and climate**

In the Paleocene interval at Site M0077, carbonate deposition dominates, and palynomorphs are nearly absent, probably due to poor preservation of organic material (Lowery et al., 2018, 2020). Low values of TOC/TN (<4) observed in the Paleocene section are also an indication of degradation of organic matter, the breakdown of nitrogenous compounds to ammonia, and subsequent $CO_2$ release via oxidation (Müller, 1977; Meyers and Shaw, 1996). The Late Paleocene palynological samples in the carbonate hardground represent the oldest dinoflagellate assemblages observed in abundances sufficient for

paleoecological interpretation. Dinoflagellate cyst and dinosterane concentrations peak in the early PETM interval, then decrease through the rest of the recovered PETM, suggesting that the extreme warmth during the PETM resulted in heat-stressed plankton within the Chicxulub impact crater, similar to the eastern equatorial Atlantic (Frieling et al., 2018). Dinoflagellate assemblages record a peak in *Apectodinium* relative abundance in the lowermost PETM sample and then a decrease in abundance through the rest of the PETM. Increases in the relative abundance of Goniodomidae through the

PETM likely indicate an intensification in salinity stratification (e.g., Frieling and Sluijs, 2018). In the later Ypresian dinoflagellate assemblages, *Spiniferites* becomes the dominant genus, and *Apectodinium* are nearly absent. The PETM nannoplankton assemblage contains malformed *Discoaster* specimens, which may represent ecophenotypes that migrated to a deep photic zone refuge to escape inhospitable SSTs and became malformed due to increased organic matter remineralization and calcite undersaturation (Bralower and Self-Trail, 2016).

A notable acme of fungal spores occurs in the middle part of the PETM. This acme is dominated by aff. *Nigrospora* sp., possibly suggesting increased moisture levels, which resulted in increased fungal decomposition of herbaceous and woody substrates on land (Dighton, 2016; Wang et al., 2017) as well as increased terrestrial runoff. The release of soluble nutrients by saprotrophic fungi such as *Nigrospora* may have significantly contributed to increased marine productivity at Site M0077 during the PETM. However, *Nigrospora* can also be transported by dust storms (Wang et al., 2017), and lives in marine environments (Dighton and White, 2017), including in deep-sea sediments (Singh et al., 2012) and microbial mats in anoxic, hypersaline coastal environments (Cantrell et al., 2006).

The PETM pollen and plant spore assemblage is broadly similar to later Ypresian assemblages observed higher in the core, with angiosperm pollen dominant, particularly reticulate tricolpate/tricolporate pollen of unknown lower botanical affinity (e.g., *Fraxinoipollenites* spp. and *Retitricolporites* spp.), *Malvacipollis* spp. (Euphorbiaceae), *Psilatricolpites* sp. A, and *Ulmipollenites krempii* (Ulmaceae). The PETM pollen and plant spore assemblage is distinguished from the later Ypresian assemblages by higher relative abundances of *Boehlensipollis* sp. A (Elaeagnaceae), *Malvacipollis* spp., and *Scabratricolpites* sp. A (Smith et al., 2020a, 2020b), suggesting that these may be thermophilic taxa. Lower plant spores and gymnosperm pollen are rare in both the PETM and later Ypresian assemblages. The main pollen source area is interpreted as a lowland tropical forest and shrubland (Smith et al., 2020a, 2020b). Pollen with affinity to the Pinopsida and Ulmaceae may represent a contribution from more upland pollen source areas, based on their modern distributions in México and Central America. High concentrations of pollen in two PETM samples argue for a proximal pollen source area from low elevation carbonate terrain in the Yucatán Peninsula, consistent with modeled prevailing surface currents and summer wind fields from the south (Fig. 2) (Winguth et al., 2010). Globally, plant floras indicate shifts in ranges and relative abundances with low rates of extinction (Wing and Currano, 2013). These shifts are broadly indicative of warming during the PETM. Although plant assemblages in midlatitude continental interiors suggest drying during the PETM (e.g., Wing et al., 2005), PETM floral records from tropical South America (Jaramillo et al., 2010) and India (e.g., Prasad et al., 2018) suggest high levels of precipitation, whereas in tropical East Africa (Handley et al., 2012) evidence suggests a decrease in overall humidity but an increase in the intensity of precipitation events. The proxy data from Site M0077 indicate that increased temperature and humidity in the Yucatán Peninsula region during the PETM resulted in increased terrestrial input.

## 5. Conclusions

The PETM in the Chicxulub impact crater was a time of extremely high SSTs (~37.8 °C), increased terrestrial input, high surface productivity, water column stratification, and bottom water hypoxia/anoxia, with evidence for photic zone euxinia in the bottom of the section. The observed increase in terrestrial input is likely the result of increased weathering and fluvial discharge due to moist, hyperthermal conditions. This explanation is consistent with global evidence of sea level rise during the PETM. Seafloor anoxia decimated the marine benthos during the PETM, while high SSTs caused heat stress in the dinoflagellate and likely other phytoplankton assemblages. In contrast, the pollen and spore assemblage indicates the presence of a proximal humid landmass with a diverse tropical shrubby forest, which produced relatively high abundances of

Euphorbiaceae pollen. These results, in combination with previously described tropical PETM floral assemblages (Jaramillo et al. 2010; Srivastava and Prasad, 2015; Prasad et al., 2018), demonstrate that tropical vegetation was highly resilient to hyperthermal conditions.

**Data availability**

All data and supplementary methods are included as supplementary materials.

**Team list (IODP Expedition 364 Scientists)**

Joanna Morgan [Department of Earth Science and Engineering, Imperial College London, UK], Sean P.S. Gulick [Department of Geological Sciences, Jackson School of Geosciences, University of Texas at Austin, Austin, TX 78712, USA; Institute for Geophysics, Jackson School of Geosciences, University of Texas at Austin, Austin, TX 78712, USA], Claire Mellett [British Geological Survey, The Lyell Center, UK], Johanna Lofi [CNRS, Aix-Marseille Univ, IRD, Coll France; INRAE, CEREGE, Aix-en-Provence, France], Elise Chenot [GeoRessources, Université de Lorraine, CNRS, France], Gail Christeson [Institute for Geophysics, Jackson School of Geosciences, University of Texas at Austin, Austin, TX 78712, USA], Phillippe Clayes [Analytical, Environmental, and Geo-Chemistry, Vrije Universiteit Brussel, Belgium], Charles Cockell [Center for Astrobiology, School of Physics and Astronomy, University of Edinburgh, UK], Marco Coolen [Department of Chemistry, Western Australian Organic & Isotope Geochemistry Centre (WA-OIGC), Curtin University, Australia], Ludovic Ferrière [Natural History Museum, Austria], Catalina Gebhardt [Alfred Wegener Institute Helmholtz Centre of Polar and Marine Research, Germany], Kazuhisa Goto [International Research Institute of Disaster Science, Tohoku University, Japan], Heather Jones [Department of Geosciences, Pennsylvania State University, University Park, PA 16801, USA], David Kring [Lunar and Planetary Institute, Houston, TX 77058, USA], Christopher Lowery [Department of Geological Sciences, Jackson School of Geosciences, University of Texas at Austin, TX 78712, USA], Rubén Ocampo-Torres [Groupe de Physico-Chimie de l'Atmosphère, L'Institut de chimie et procédés pour l'énergie, l'environnement et la santé (ICPEES), France], Ligia Perez-Cruz [Instituto de Geofísica, Universidad Nacional Autónoma De México, México], Annemarie E. Pickersgill [School of Geographical and Earth Sciences, University of Glasgow, UK], Michael Poelchau [Department of Geology, University of Freiburg, Germany], Auriol Rae [Department of Earth Science and Engineering, Imperial College London, UK], Cornelia Rasmussen [Institute for Geophysics, Jackson School of Geosciences, University of Texas at Austin, USA], Mario Rebolledo-Vieyra [Unidad de Ciencias del Agua, Centro de Investigación Científica de Yucatán, A.C., México], Ulrich Riller [Institut für Geologie, Universität Hamburg, Germany], Honami Sato [Japan Agency for Marine-Earth Science and Technology, Japan], Jan Smit [Faculty of Earth and Life Sciences FALW, Vrije Universiteit Amsterdam, Netherlands], Sonia Tikoo [Earth and Planetary Sciences, Rutgers University, New Brunswick, NJ 08854, USA], Naotaka Tomioka [Kochi Institute for Core Sample Research, Japan Agency for Marine-Earth Science and Technology, Japan], Michael Whalen [Department of Geosciences, University of Alaska Fairbanks, Fairbanks, AK 99775, USA], Axel Wittmann [LeRoy Eyring Center for Solid State Science, Arizona State University, AZ 85281, USA], Kosei

Yamaguchi [Department of Chemistry, Toho University, Japan], Long Xiao [School of Earth Sciences, Planetary Science Institute, China University of Geosciences (Wuhan), China], William Zylberman [CNRS, L'Institut de recherche pour le développement, Coll France, Aix Marseille University, France].

## Author contributions

VS led the writing and organization of the manuscript. VS, SW, NNO, and JMO analyzed the terrestrial palynology. JV
analyzed the dinoflagellate assemblages. KG, BS, TB, and LS provided biomarker data and interpretation. MTW provided carbon and nitrogen isotopes and sedimentologic evaluation of the core. KO provided additional isotope and geochemical data. IA, JAA, and CL researched the foraminiferal assemblages. EC provided clay mineralogy data. HJ provided nannofossil data. FJR provided ichnological data. JG and FD provided magnetic data. SPG, TB, JL, JM, and other co-authors assisted with conceptualization and writing of the manuscript.

**Competing interests**

The authors declare that they have no conflict of interest.

## Acknowledgements

This research used samples and data provided by the International Ocean Discovery Program (IODP). Funding was provided by the CENEX (Center for Excellence in Palynology) Endowed Chair Fund, a 2018 James M. and Thomas J.M. Schopf
Award Student Research Grant from the Paleontological Society, an ARC-Discovery grant (DP180100982) from the Australian Research Council (ARC), a postgraduate award from Curtin University, IODP-France, the Research Foundation Flanders (FWO grant 12Z6618N), NSF OCE grants 1736951, 1737199, and 1737351, and NERC grant NE/P005217/1. Thanks to Roger E. Summons and Xingqian Cui (MIT, US) for MRM analyses. This is University of Texas Institute for Geophysics Contribution #3676 and Center for Planetary Systems Habitability Contribution #0013.

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

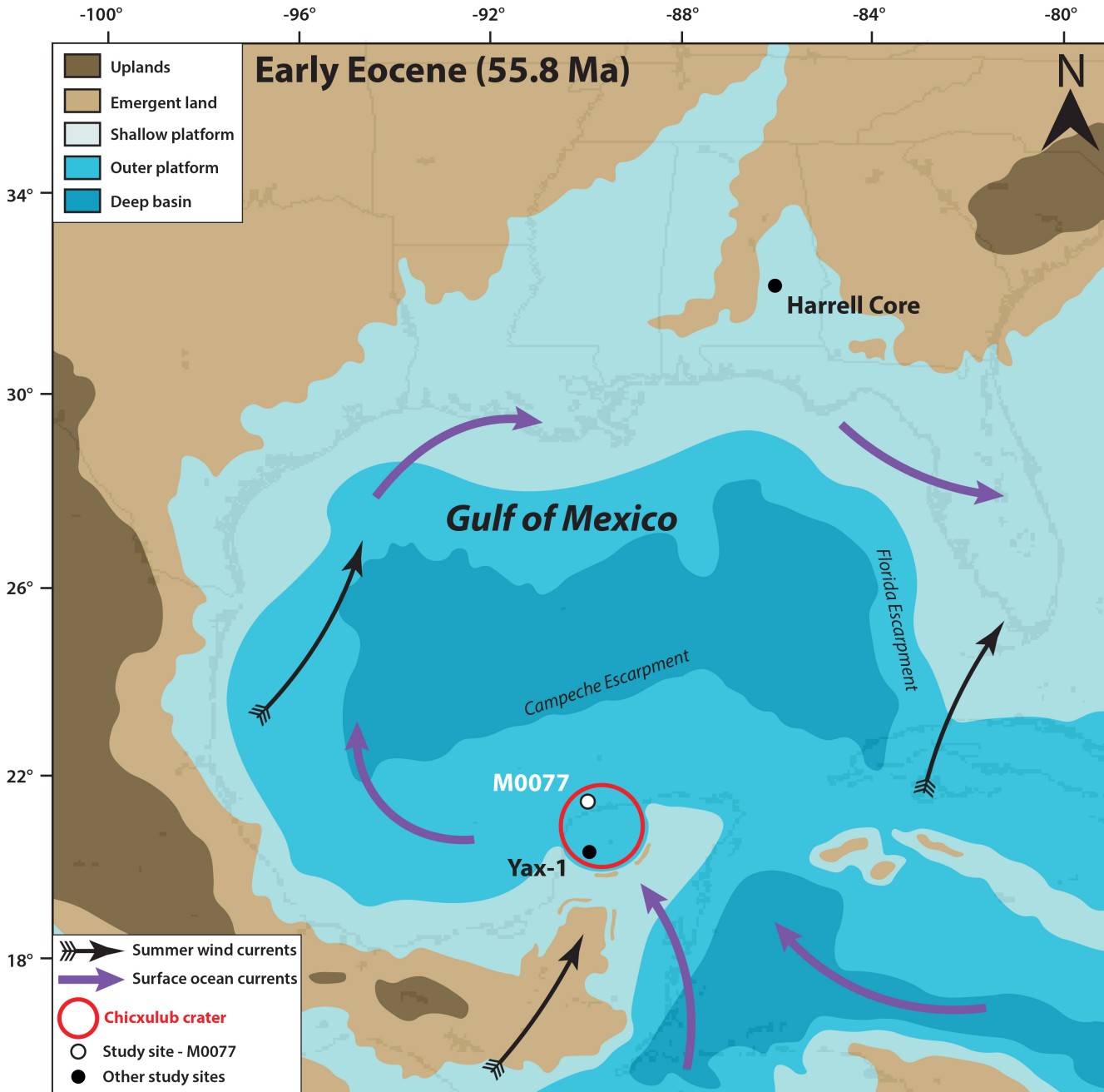

Figure 1. Paleocene-Eocene Thermal Maximum (55.8 Ma) paleogeography of the Gulf of Mexico and surrounding regions, modified from Scotese and Wright (2018), with locations of Site M0077 (IODP 364), Yax-1 (Whalen et al., 2013), and the Harrell Core in east-central Mississippi (Sluijs et al., 2014). The Harrell Core location has been adjusted to match the paleo-latitude/longitude at the PETM. Surface ocean currents and summer wind fields from Winguth et al. (2010).

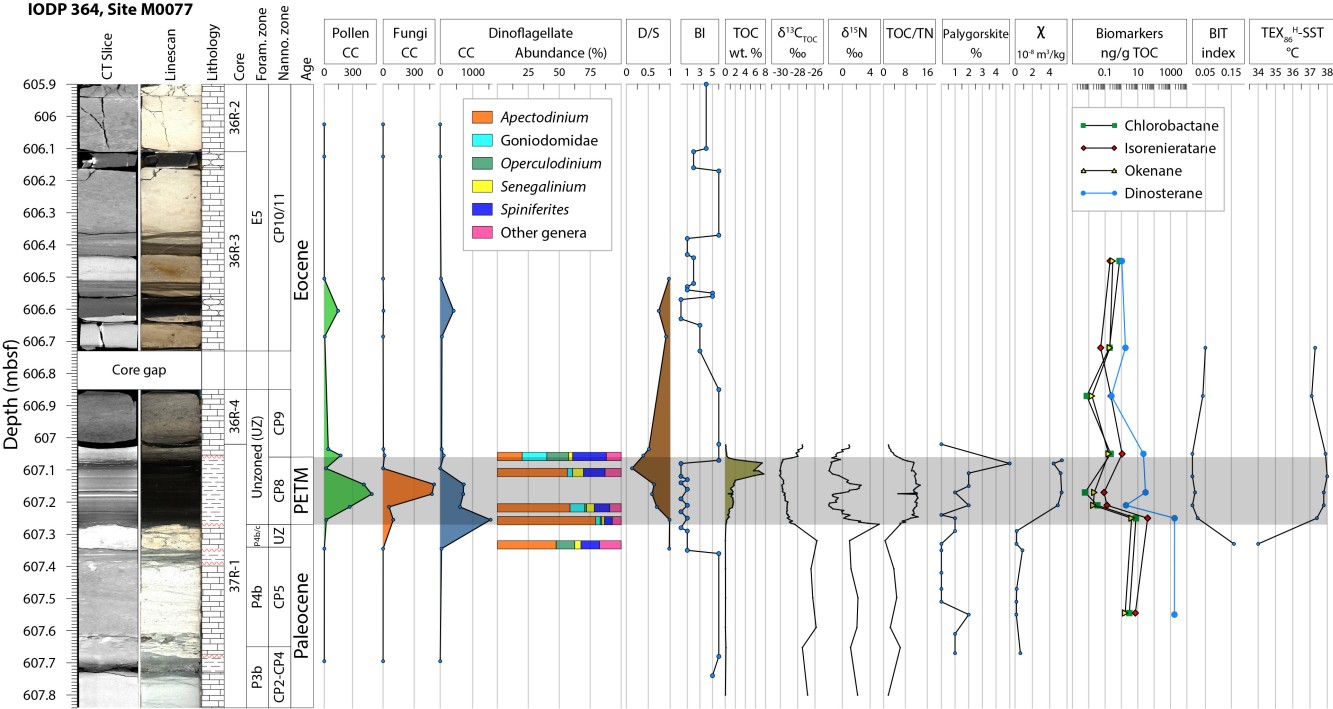

**Figure 2. Stratigraphic column of Site M0077.** Palynological concentrations (CC) are given as specimens/gram. $\delta^{15}N$ is reported relative to atmospheric $N_2$, $\delta^{13}C_{TOC}$ is reported relative to VPDB. BI=bioturbation index (Taylor and Goldring, 1993), CT=computed tomography, D/S=dinoflagellate cyst to pollen and plant spore ratio (Warny et al., 2003), TOC/TN=total organic carbon/total nitrogen, TOC=total organic carbon, $\chi$=magnetic susceptibility. Lithological symbols: rectangular blocks=limestone, dashes=shale/claystone, obround ovals=chert, red wavy lines=unconformities.

