# Peer review of "Life and death in the Chicxulub impact crater: A record of the Paleocene-Eocene Thermal Maximum"

_Climate of the Past, 2020_

## Referee Comment (RC1) · C. Jaramillo (Referee) · 28 May 2020

All PETM papers are interesting by default as this is an unique time interval that it is not found easily. Smith et al present the PETM record of a core drilled at the Chicxulub crater. The manuscript presents enough data that support that the PETM is present indeed and mainly focuses on the palynological content of the PETM including both marine and terrestrial palynomorphs. The dinoflagellate record shows the classical *Apectodinium* spike that is seen at the PETM across the world, mainly in tropical settings. The terrestrial record shows an abundance of fungal spores and a pollen-spore assemblage that is a mixture of tropical and mid-temperate taxa, that is expected given

the paleolatitude of the site. There is no evidence of a collapse of the vegetation during or following the PETM. Unfortunately, the late Paleocene record is missing as organic matter did not preserved, therefore, it is impossible to evaluate the actual change on late Paleocene floras due to the PETM. Nevertheless, this is the only PETM of the Gulf Coast-northern Caribbean region and a valuable addition to our global understanding of the PETM that is in need of many more sites, in particular those with a terrestrial signal. Paper is short, well written and of interest for the broad audience of CP.

I have several comments on specific items as follow:

1. Because $\delta$13C values of bulk sediments can be affected by the total organic carbon of a sample (Wing et al 2005 Science); you need to apply the Wing residuals method to the $\delta$13C record. And in this site, this seems to be evident. The discussion, then, needs to follow the pattern of the residual rather than the raw $\delta$13C values.

2. Line 185: include the TEX of the marine core in Jaramillo et al 2010 (28-31C for the late paleocene to 31-34 for the early Eocene), that is useful to take into account as the early eocene temperature in your core is slightly cooler than the PETM. Furthermore, the core is the closest to your site with tropical TEX86 marine data (caribbean of Colombia). I feel awkward asking you to use data from one my papers, but you cited the paper already and perhaps missed the TEX86 data.

3. Line 235: relating the high abundance of fungal spores to humid forest and grasslands is a large assumption. It could be a simple taphonomic effect as fungal spore wall are often thicker than pollen walls and tend to preserve better in deep waters and/or when rates of weathering are higher. Can you discard fungal spores being a taphonomic signal?

4. Line 240: grasses are also very common in aquatic settings (e.g. floating grass-islands in amazonas river), and aquatic grasses are probably the source of low quantities of grass pollen rather than savannas. Extensive savannas are only seen in Miocene and younger strata, therefore, there is a need for additional evidence if you

are proposing savannas at the PETM, even more even more when your "grass" record consists of a single grain in a single sample (607.1 m) 5. Liliacitides variegatus in the PETM? That is hard to believe. It must a reworked taxon as it became extinct in the Cenomanian

Hope to see it published soon Carlos Jaramillo

---

## Referee Comment (RC2) · Kate Littler (Referee) · 27 Jun 2020

General comments:

The authors present a new partial PETM section from the Gulf of Mexico, recovered from Site M0077, IODP Expedition 364. The ∼body of the PETM is identified in this core on the basis of multiple lines of biotic (e.g. calcareous nannofossil, planktic foraminiferal) and geochemical evidence (e.g. a 4 per mille shift in d13CTOC), and appears to be robust, albeit incomplete and bracketed with significant hiatuses of >1 myr. The authors have applied a multi-proxy approach including palynology and organic geochemistry to shed light on both the marine and terrestrial expression of this

global event in this region.

This study presents new multi-proxy data on the ∼body of the PETM from an under-studied region, and so is a valuable contribution to our understanding of this enigmatic event. It appears the palynological data have already been published in terms of the species/genera present (Smith et al., 2019; 2020), but the relative abundance data and the geochemical data are new. We have a particular dearth of information on the terrestrial impact of the PETM in this region, so this contribution is welcome. The lack of late Paleocene data predating the event is a shame, and so the study does lack some temporal context, but sometimes the lithologies in question just don't play ball with our proxies. The TEX86 data needs more careful treatment and exploration of the caveats. Overall, I think the paper will be of interest to the readership of COTP and the Palaeogene community more specifically.

The manuscript is concise and clear, with a high standard of writing throughout. There are minimal typos, and instances of poor syntax and grammar are rare. The two figures are clearly drafted and well captioned. The paper is well referenced and the bibliography is largely complete.

Specific comments:

1. The Methods section is a bit skimpy and should be expanded, especially considering there is no extra methodological detail in the Supplementary Information (just the data tables xlsx file – unless I am missing something?).

- More needs to be said regarding the overall sampling strategy of the samples taken for geochemistry and palynology, and the resultant resolution of your samples (both in terms of depth and time). "N= 51 samples" is not sufficient.

- Line 71: Further elaboration on the Bioturbation Index would be welcome, e.g., what defines more or less bioturbation in a section? What does a score of 1 entail vs. a score of 6? When looking at Figure 2 I assume a higher number means more bioturbation,

but we shouldn't have to go to Taylor and Goldring, (1993) to check.

-Line 75: Please state at which institution the ECS and Delta instruments are located. It is also typical to state which external (and internal) standards are used to calibrate the isotopic measurements. Is the instrument precision you quote to one or two sigma? Please note that $\delta$13CTOC data is reported relative to VPDB and $\delta$15N is reported relative to atmospheric N2, (as is stated in the caption of Fig. 2 but not here in the Methods).

-What instruments did you use to measure the biomarker data with (GC-MS. . . HPLC-MS. . .) and where were these located? How did you extract the samples? What is the $\pm$ error on the TEX86 measurements (incorporating both the analytical and calibration error)? You should state that the GDGT distributions were also used to construct the BIT Index data.

2. Broader temporal context: 2a. The lack of Paleocene palynological and geochemical data is challenging (but not the authors' fault) as it removes some of the context for the PETM. For example, one is left wondering if the elevated temperatures or the distributions of palynological components is unique to the "event" or is merely similar to the background Palaeocene-Eocene signature of the region? I note that the palynology data presented in the Supplementary Info file stretches way up into the early Eocene (up to 505.88 mbsf, ~48 Ma). I realise the focus of this MS is on the PETM but have you considered presenting and discussing this Eocene data in this MS to give better context to your PETM data? I know much of this data is already published as part of Smith et al., 2019 and 2020, but the data in these publications is presented as taxonomic reports (illustrated in the form of plates) and is not graphed up in the same manner as you have done for the PETM data here. This context would be welcome and would help to support statements in your MS such as "The PETM pollen and plant spore assemblage is broadly similar to later Ypresian assemblages observed higher in the core, with angiosperm pollen dominant and rare lower plant spores and gymnosperm pollen." (line 235), for which you provide no supporting reference or data at

present.

2b. I'd like to see more discussion of whether the "PETM black shale" is unusual in the overall context of the Paleocene-Eocene strata at this site, or are there many higher TOC intervals? I see from Fig 2 that there is another little black shale at 606.6 mbsf and another at 606.15 mbsf. These apparently sit within E5 and CP10/11 so are likely to be within ∼EECO (rather than being ETM-2 or -3)- are there any others, and if so what does this mean for the oceanographic conditions of the region with time? Are the low oxygen, low bioturbation conditions during the PETM here unique or not?

3. Sedimentology and stratigraphy: 3a. There is good discussion of the sedimentology of this part of the core in section 4.1., including the unconformities, but this is not then clearly annotated onto Figure 2. It takes quite a bit of reading and flicking back and forth between the figure and text to work out which packstone or hardground you're talking about, so these could be linked more clearly using specific mbsf and arrows/ annotations on the figure. Perhaps a graphic log would help too? In particular I don't think the major hiatuses surrounding the PETM are signposted clearly enough in Figure 2. In <10 cm around 607.3 mbsf you jump from CP5 to CP8, which is about 2 million years of time at the minimum. The way the data is presented in the graphs this is not immediately clear.

3b. On the basis of both the sedimentological, geochemical and palaeontological data, (and by your own omission in lines 159-167) there is only a partial record of the PETM preserved in this core. It likely represents either the onset and body or just the body of the event. I would recommend you therefore use "partial PETM" and/or "body of the PETM" throughout the MS to make this clear to readers.

4. The limitations of the TEX86 data need a bit more discussion and exploration. -In addition to the TEX86H calibration (Kim et al., 2010) it would be a good idea to also calibrate the TEX86 data using the BAYSPAR calibration (Tierney and Tingley, 2015. Scientific Data), which would yield broadly the same trends in SST but with different

absolute values. This may change some of your interpretations with relation to heat stress and the tolerance of marine organisms during the PETM. -In the Supplementary Information I would recommend that you present the raw GDGT abundance data (not just the TEX86 ratio), because others can then recalibrate your data when, inevitably, a new calibration comes along. - In order to have a bit more confidence in your TEX86 data (and SSTs) it's a good idea to apply the series of "tests" to the data. These include the Methane Index, %GDGT-0 index etc… (see Hollis et al., 2019. Geosci. Model Dev. Section 4.4, for the links to the all the relevant papers and methodologies). You presumably already have all the data you need (the GDGT distributions) so it's not more analytical work. - Have these samples been checked for maturity (e.g., using the hopane distribution) which can skew TEX86 values (e.g., Schouten et al., 2004. Organic Geochemistry)? Indeed, considering the samples lie at >600 mbsf in the section is there any evidence for diagenetic alteration?

5. I find it interesting that the purple sulphur bacteria markers (e.g. isorenieratane) are apparently detectable throughout the sequence including in the late Paleocene carbonates, not just in the black shale interval. Can you comment on this in terms of the oceanographic conditions at this time? Was some degree of euxenia common within this crater basin over the long-term?

6. In the Supplementary Info data file ("Pollen and Spore Counts" tab) the data is also plotted against absolute age. Please either detail the age model used to calculate these ages (in the MS) or remove the column. It may seem petty but when other people come to re-plot your data in their own publications it may get copied over without further scrutiny.

I look forward to seeing the final corrected paper published. Best wishes, Kate Littler.

---

## Author Response (AR1)

Dear Dr. Reyes,

Many thanks to both you and the two peer reviewers for your time and your helpful suggestions for improvement of this manuscript. We have addressed the concerns of the two peer reviewers in our previously uploaded responses. These two responses have been included with this document, along with a marked-up version of the manuscript, to provide a point-by-point, detailed response. We have added the raw GDGT abundance data as well as supplementary methods to our supplementary materials in a ZIP file. Figure 2 has been revised to include a lithological column in response to Dr. Littler's point 3a. We have added more discussion of the Ypresian section of the palynological record, specifically in lines 260-261 and 272-278. The terrestrial pollen and spore record exhibits less dramatic assemblage changes between the PETM and later Ypresian than the fungal and dinoflagellate records, with higher abundances of some possibly thermophilic pollen taxa in the PETM assemblages. It has been a pleasure working with you and the other peer reviewers, and we hope you find that the revised manuscript addresses your suggestions.

Sincerely,

Vann Smith

**Response to Carlos Jaramillo:**

Many thanks for your comment, Dr. Jaramillo! We will respond to your points in order.

1. Wing et al. (2005) calculated anomaly values of $\delta^{13}C_{org}$ based on a logarithmic regression of $\delta^{13}C_{org}$ versus wt. % $C_{org}$ in terrestrial paleosols. These anomalies were given to account for isotopic enrichment by soil microbes. However, in our marine section, the carbon isotope signature is also affected by the proportion of terrestrial versus organic matter, as well as other factors, and we do not think that the relative contribution of these variables can not be deconvolved into anomalies based on a simple linear regression.

2. We are happy to cite the $TEX_{86}$ data from Jaramillo et al. (2010), this is an important comparison.

3. Not only is the relative abundance of fungal spores versus pollen higher in the PETM samples than all other samples in the Site M0077A core, the absolute concentration (specimens/gram) of fungal spores is higher than all other samples, including samples in the later Ypresian section near the top of the core with much higher pollen and plant spore concentrations (over 10,000 grains/gram at some depths) than the PETM samples. Taphonomic processes which would preferentially preserve fungal spores compared to pollen would increase the relative abundance of fungal spores in the palynomorph assemblage but would not explain the higher concentrations of fungal spores relative to later Ypresian samples with excellent preservation. This has been clarified in the results section, and additional discussion about the possible paleoecologies of the aff. *Nigrospora* spores has been provided. Elsewhere in the Gulf Coast (Demchuk, Denison, and O'Keefe, unpublished data) higher concentrations of fungal spores in terrestrial and nearshore PETM sections have been observed along with near static levels of terrestrial palynomorphs; of the fungal spores, aff. *Nigrospora* sp. are most common. This suggests two things: 1) that there were increased moisture levels at the time, which A) increased decomposition, (Wang et al. 2017; Dighton 2016) and B) resulted in increased runoff, thus bringing palynomorphs into the Gulf of Mexico; and 2) that increases in terrestrial run-off may have led to higher productivity noted in overlying samples because the actions of saprotrophic fungi result in the release of soluble nutrients into the environment that would otherwise be immobilized in plant tissues (Dighton 2016).

4. Your point is well taken. The sentence "Low abundances of grass pollen (*Monoporopollenites annulatus*) in the PETM suggest a minor grasslands component of the flora" has been removed from the text. References to "grassland" have been removed. The presence of only a single specimen of probable grass pollen in the PETM assemblage indicates that extensive grass cover was not present in the pollen source area.

5. The range given for *Liliacidites variegatus* in the original description by Couper (1953) is Upper Cretaceous to Lower Oligocene for New Zealand. Elsik (1968) observed *L. variegatus* in the Paleocene of Texas, and Rouse and Matthews (1988) observed *L. variegatus* in the Eocene of British Columbia. Additional Cenozoic occurrences of this species can be found in Palynodata Inc. and White (2008), or online at paleobotany.ru.

Couper, R. A. (1953). *Upper Mesozoic and Cainozoic spores and pollen grains from New Zealand* (Vol. 22). Alexander Doweld.

Dighton, J. (2016). *Fungi in ecosystem processes* (Vol. 31). CRC press.

Elsik, W. C. (1968). Palynology of a Paleocene Rockdale lignite, Milam county, Texas. I. *Morphology and taxonomy. Pollen & Spores*, *10*, 263-314.

Jaramillo, C., Ochoa, D., Contreras, L., Pagani, M., Carvajal-Ortiz, H., Pratt, L. M., ... & Rodriguez, G. (2010). Effects of rapid global warming at the Paleocene-Eocene boundary on neotropical vegetation. *Science*, *330*(6006), 957-961.

Palynodata, I., & White, J. M. (2008). Palynodata Datafile: 2006 version.

Rouse, G. E., & Mathews, W. H. (1988). Palynology and geochronology of Eocene beds from Cheslatta Falls and Nazko areas, central British Columbia. Canadian Journal of Earth Sciences, 25(8), 1268-1276.

Wang, M., Liu, F., Crous, P. W., & Cai, L. (2017). Phylogenetic reassessment of Nigrospora: ubiquitous endophytes, plant and human pathogens. *Persoonia: Molecular Phylogeny and Evolution of Fungi*, *39*, 118.

Wing, S. L., Harrington, G. J., Smith, F. A., Bloch, J. I., Boyer, D. M., & Freeman, K. H. (2005). Transient floral change and rapid global warming at the Paleocene-Eocene boundary. *Science*, *310*(5750), 993-996.

**Response to Kate Littler:**

Many thanks for your comment, Dr. Littler! We will respond to your points in order.

1. We have added a supplementary Methods document in the Supplementary Information. A short discussion of the sampling strategy and resolution has been added to the Methods section in the main manuscript. In some cases the sampling resolution was limited by practical considerations (e.g., lack of funding). The Bioturbation Index has been briefly described. The location of the ECS and Delta instruments, and the analytical precision, has been included in the Supplementary Information. A mention of the $\delta^{15}N$ and $\delta^{13}C_{TOC}$ isotope standards has been added to the Methods section. Description of the methods used for biomarker analysis is included in the Supplementary Information.

2a. As you note, the Paleocene record has been challenging due to extremely low palynological abundances and low TOC. Although multiple late Paleocene samples were analyzed for biomarkers, TOC was too low to determine $TEX_{86}$ for all but one sample. The latter point has been clarified in the Methods section. Also, there are at least two manuscripts in preparation by various co-authors which deal with later Eocene hyperthermals and the Early Eocene Climatic Optimum, limiting our ability to discuss these upcoming results. However, more information on the palynological assemblages in the PETM section relative to the later Ypresian has been added.

2b. TOC values in the Site M0077A core generally increase upsection (Gulick et al., 2017). Other laminated dark shale and marlstone sections are present in the later Ypresian, notably a laminated marlstone section at ~598-597 mbsf. This section is the subject of current research, so we are limited in our ability to discuss these results. The two black layers visible above the partial PETM section in Figure 2 are actually black cherts; this has been clarified near the beginning of the Results section.

3a. Figure 2 has been modified to include a separate column illustrating the unconformities present. Abbreviations have been used to label lithological units in the stratigraphic column. Additionally, the first paragraph of section 4.1 has been revised to include depth ranges for the lithological units described.

3b. In the abstract the PETM record is now referred to as "…a new record of the body of the PETM." The incomplete nature of the PETM record at this site is discussed in the Results section (which has been reorganized to include some text previously in the Discussion section). The Discussion section now begins: "As described earlier, the PETM section in the Site M0077 core is bracketed by unconformities and incomplete, with the onset and recovery missing, and only part of the body of the PETM preserved." After discussion with the co-authors, we considered that referring to the PETM record throughout the manuscript as a "partial PETM record" is unnecessary, as we explain the incompleteness of the record in the manuscript.

4. The BAYSPAR and linear TEX86 calibrations yield unrealistically high PETM SSTs in excess of 44 °C, likely above the heat tolerance for dinoflagellates, foraminifera, and other eukaryotic plankton. The $TEX_{86}^{H}$ calibration of Kim et al. (2010) provides more realistic SST estimates which are in agreement with other published GDGT data for the PETM in the region (Zachos et al., 2006; Jaramillo et al., 2010, Sluijs

et al., 2014). A short discussion of this has been added to the manuscript. Complementary data comprising BIT, MI and $f_{Cren}$ to evaluate applicability of the $TEX_{86}$ proxy (exclusion criteria as compiled in O`Brian et al. (2017)) are provided in the supplementary materials. The thermal maturity as determined by side chain isomerization of $C_{29}\alpha\alpha\alpha$ steranes [20S/(20S+20R)] and $C_{31}\alpha\beta$ hopanes [22S/(22S+22R)] average 0.13 and 0.35, respectively (see supplementary materials), which is indicative of a low maturity equivalent to a vitrinite reflectivity of 0.3 to 0.35%. This is supported by Rock Eval Tmax values averaging 428°C. No maturity impact on the GDGT data is observed. Preservation of immature biomarkers is further supported by the presence of thermally labile aromatic carotenoids. The first paragraph of section 4.2 has been revised to include discussion along these lines.

5. As described, the carotenoid biomarkers are present, albeit in trace concentrations, above and below the black shale interval. In the crater basin, evidence of periods of photic zone euxinia was reported for the limestone interval prior to the PETM, as shown by Schaefer et al. (2020). Here, different sources (microbial mats versus open water column PZE) for the elevated carotenoids have been proposed. Abundant PZE markers are ascribed to plankton concentrated at the chemocline, as found in restricted marine basins where high concentrations of hydrogen sulphide occur within the sunlight zone. Alternatively, PZE markers reflect a change in the microbial community, either within the water column triggered by stratification, or via the transport of microbial mats from the shallow waters surrounding the crater, as indicated by elevated concentrations of cyanobacterial biomarkers and intact heterocyst glycolipids.

6. The absolute age column has been removed.

Gulick, S., Morgan, J., Mellett, C. L., Green, S. L., Bralower, T., Chenot, E., Christeson, G., Claeys, P., Cockell, C., Coolen, M. J. L., Ferrière, L., Gebhardt, C., Goto, K., Jones, H., Kring, D., Lofi, J., Lowery, C., Ocampo-Torres, R., Perez-Cruz, L., … Zylberman, W. (2017). Site M0077: Post-Impact Sedimentary Rocks. In *Chicxulub: Drilling the K-Pg Impact Crater* (pp. 1–35). International Ocean Discovery Program.

Jaramillo, C., Ochoa, D., Contreras, L., Pagani, M., Carvajal-Ortiz, H., Pratt, L. M., ... & Rodriguez, G. (2010). Effects of rapid global warming at the Paleocene-Eocene boundary on neotropical vegetation. *Science*, *330*(6006), 957-961.

O'Brien, C. L., Robinson, S. A., Pancost, R. D., Damsté, J. S. S., Schouten, S., Lunt, D. J., ... & Farnsworth, A. (2017). Cretaceous sea-surface temperature evolution: Constraints from TEX86 and planktonic foraminiferal oxygen isotopes. *Earth-Science Reviews*, *172*, 224-247.

Schaefer, B., Grice, K., Coolen, M. J., Summons, R. E., Cui, X., Bauersachs, T., ... & Freeman, K. H. (2020). Microbial life in the nascent Chicxulub crater. *Geology*, 48(4), 328-332.

[revised manuscript text omitted]

---

## Author Response (AR2)

Dear Dr. Reyes,

No changes have been made to the manuscript since the last revision. Many thanks for your time.

Sincerely,

Vann Smith